# Serum β-Defensin 2, A Novel Biomarker for the Diagnosis of Acute Infections

**DOI:** 10.3390/diagnostics13111885

**Published:** 2023-05-28

**Authors:** John G. Routsias, Dionysia Marinou, Maria Mavrouli, Athanasios Tsakris, Vassiliki Pitiriga

**Affiliations:** Department of Microbiology, School of Health Sciences, National and Kapodistrian University of Athens, Mikras Asias 75, 11527 Athens, Attica, Greece

**Keywords:** hBD2, defensins, infection, procalcitonin, CRP, biomarker

## Abstract

Background: Defensins are natural antimicrobial peptides that the human body secretes to protect itself from an infection. Thus, they are ideal molecules to serve as biomarkers for infection. This study was conducted to evaluate the levels of human β-defensins in patients with inflammation. Methods: CRP, hBD2 and procalcitonin were measured in 423 sera of 114 patients with inflammation and healthy individuals using nephelometry and commercial ELISA assays. Results: Levels of hBD2 in the serum of patients with an infection were markedly elevated compared to those of hBD2 in patients with inflammation of non-infectious etiology (*p* < 0.0001, t = 10.17) and healthy individuals. ROC analysis demonstrated that hBD2 showed the highest detection performance for infection (AUC 0.897; *p* < 0.001) followed by PCT (AUC 0.576; *p* = ns) and CRP (AUC 0.517; *p* = ns). In addition, analysis of hBD2 and CRP in patients’ sera collected at different time points showed that hBD2 levels could help differentiate inflammation of infectious and non-infectious etiology during the first 5 days of hospitalization, while CRP levels could not. Conclusions: hBD2 has the potential to serve as a diagnostic biomarker for infection. In addition, the levels of hBD2 may reflect the efficacy of antibiotic treatment.

## 1. Introduction

Inflammation is a defense mechanism of the immune system which aims to maintain tissue homeostasis during infection and tissue injury. The entire process of the inflammatory response is mediated by a variety of regulators involved in the selective expression of pro-inflammatory molecules, such as inflammatory cytokines (e.g., IL-6).

Antimicrobial peptides (AMPs) play a critical role in the innate immune system, as they fight off infection and provide protection to the host [1]. They are natural peptide antibiotics produced by the human body as a response to infection [2,3]. A wide range of AMPs has been detected in epithelial cells derived from the skin as well as the respiratory, intestinal and reproductive tracts [4]. They constitute a diverse group of small peptides, and are classified into several categories based on their primary structures and topologies [2,4,5,6,7,8]. The most important peptide families among them are defensins, cathelicidins and REG3 lectins [1,9].

Defensins represent an evolutionary ancient family of antimicrobial peptides with pronounced antimicrobial activity [10,11,12]. They are cationic, cysteine-rich and amphipathic polypeptides consisting of 28–42 amino acids. They possess a conserved structural fold, containing six highly conserved cysteine residues, which form three pairs of intramolecular disulfide bonds [13,14,15]. Defensins have been detected in a variety of organisms, including plants, insects and invertebrate and vertebrate animals [8,16,17]. Based on the spatial arrangement of the cysteine residues and the topology of the disulfide bonds, the defensins are classified, in humans, as α-defensins, β-defensins and θ-defensins [17,18].

α-Defensins are mainly found within the azurophilic granules of the neutrophils and have antimicrobial activity [19,20], while β-defensins are mainly produced by various types of epithelial cells [21]. Defensins may be induced or expressed constitutively. Human monocytes [22] and NK cells [23] are constantly and continuously expressing the myeloid α-defensins HNP 1-3 [24]. On the contrary, the expression levels of β-defensins are transcriptionally induced in various tissues in association with the response to the production of pro-inflammatory cytokines.

Defensins play an important role in the induction of immune responses against microbial and viral infections. They are involved in adaptive immunity, signal transduction, regulation of the inflammatory effects and wound repair. These polypeptides also participate in the expression of cytokines and chemokines, chemotaxis, control proliferation, production of histamine and enhancement of antibody responses [25,26,27,28,29]. In addition, they can induce and promote anti-tumor immunity [29]. However, the main function of defensins seems to be the elimination of bacteria and fungi. More specifically, α-defensins show microbiocidal activity against Gram-positive and Gram-negative bacteria, fungi, spirochetes, protozoa and viruses [17]. Moreover, each β-defensin is characterized by its ability to kill or inhibit the in vitro growth of a wide range of bacteria and fungi, especially at low salt and protein plasma concentrations [17,30,31,32,33]. Human defensins may react in oligomerized forms with the negatively charged microbial membranes and create pores. These pores lead to the leakage of the cytoplasmic content and the lysis of the bacterial cell [34]. In addition, defensins show chemotactic activity, similar to that of chemokine MIP-3a.

hBD1 is primarily expressed in the urinary and respiratory tract epithelia [35,36,37]. hBD2 was originally isolated from psoriatic skin lesions and is detected mainly in the respiratory tract, skin and urinary and gastrointestinal tract epithelia [38]. hBD2 is a component of the epithelial defense system of the skin and the respiratory tract, playing a key role against infection. It is produced upon the contact of keratinocytes with Gram-negative bacteria or pro-inflammatory cytokines. It has antimicrobial activity, mainly against Gram-negative bacteria (such as *E. coli* or *Pseudomonas aeruginosa*) and Candida [38,39]. Furthermore, this peptide and its mRNA have been detected in sessional epidermal keratinocytes in both psoriasis and mastitis, as well as in the infected lung epithelia of patients with cystic fibrosis [39,40]. Like hBD1, the antimicrobial activity of hBD2 is inhibited by high concentrations of NaCl [41]. Since defensins are natural antimicrobial molecules that the human body secretes to protect itself from infection, they are ideal to study as biomarkers for the inflammation of infectious etiology. This study was aimed at examining if the levels of human β-defensins (hBD1 and hBD2) in patients with inflammation can be used as biomarkers for infection or cancer, as well as to examine their diagnostic impact with regard to other conventional biomarkers of inflammation such as CRP or PCT.

## 2. Materials and Methods

### 2.1. Study Design and Participants

We prospectively reviewed via the laboratory software system sLIS (Infomed, Ltd., Athens, Greece) all hospitalized patients in Metropolitan Hospital (Athens, Greece) over a 1.5-year period (2019 to 2020). The primary selection criterion was the existence of at least one sample of markedly elevated CRP levels (more than 10 times the normal upper limit of 0.5 mg/dL). All principal medical and surgical subspecialties, except transplant and obstetrics, are offered in Metropolitan Hospital and thus no key adult population were excluded.

Hospitalized patients with high levels of c-reactive protein (CRP) due to an inflammation (of various origins) as well as healthy individuals were recruited. Blood from peripheral veins of patients and healthy individuals was collected and centrifuged after informed consent was obtained. Serum was separated, aliquoted and stored at −80 °C until use.

In total, 423 serial sera of 114 patients with inflammation were included in the study as well as 46 healthy individuals. All hospitalized patients’ medical records were examined in detail and they were categorized upon determination of whether they had an active infection at the time of the sample collection and their current diagnosis (e.g., cancer, trauma, autoinflammatory disease, etc.). Patient sera were divided into groups: (a) without cancer or infection (including acute myocardial infarction, trauma, autoinflammatory disease patients etc), (b) with active infection but without cancer, (c) with cancer but without active infection and (d) with both cancer and active infection. Groups regarding an active infection were further subdivided according the type of infection as follows: tissue infections, urinary tract infections, gastrointestinal tract infections, respiratory tract infections and bloodstream infections (bacteraemia). All samples obtained from healthy individuals were found to have low CRP levels and below the positivity limit.

### 2.2. Laboratory Measurements

The concentrations of IL-6, procalcitonin (PCT), hBD1 and hBD2, respectively, were measured in serum using commercial enzyme-linked immunosorbent assays (ELISA) and nephelometry.

### 2.3. hIL-6, hBD1 and hBD2 ELISA

The concentrations of hIL-6, hBD1 and hBD2 were measured using ELISA development kits (Peprotech, Inc., London, UK) according to the manufacturer’s instructions. These kits contain the components required for the quantitative measurement of natural hIL-6, hBD2 and hBD2 in an ELISA sandwich format. Patient’s serum was added at a dilution of 1:2, 1:15 and 1:5 for hIL-6, hBD1 and hBD2 ELISA in order to fall within each kit’s measurement range (24–1500 pg/mL, 4–1000pg/mL and 16–2000 pg/mL, respectively). Standard curves were produced by serial dilutions of the kits’ hIL-6, hBD1 and hBD2 recombinant antigens.

### 2.4. Procalcitonin ELISA

For the measurement of human PCT, a human procalcitonin ELISA kit (RayBiotech, Peachtree Corners, GA, USA) was used following the manufacturer’s instructions. Patient’s serum was diluted 1:5. A standard curve was produced by serial dilutions of standard human PCT.

### 2.5. Nephelometric CRP Assay

C-reactive protein levels were measured using the Architect c16000 multichannel analyzer (Abbott Laboratories Inc., Irving, TX, USA).

### 2.6. Blood Count, Blood Chemistry Tests and Blood Coagulation Measurements

Blood analysis was carried out with a Beckman Coulter LH 750 Analyzer Station (Brea, CA, USA). Glucose, Urea, Creatinine, K^+^, Na^+^, TBIL, LDH, ALB and CPK were measured by a COBAS 6000 series (ROCHE Diagnostics, Hague Rd, IN, USA)) and INR and D-dimer were measured using a Siemens BCS XP system (Siemens, Tarrytown, NY, USA).

### 2.7. Statistical Analysis

Data analysis was carried out using SPSS version 19.0 software (SPSS, Chicago, IL, USA). The *t*-test method was used for comparison of patients with inflammation of infectious vs non-infectious etiology as well for comparison of patients with healthy individuals. Pearson correlation coefficient analysis was performed to determine the associations between the hBD1, hBD2, CRP, IL-6, PCT, WBC, Neu, HCT, ESR, Glucose, Urea, Creatinine, K^+^, Na^+^, TBIL, LDH, ALB, CPK, INR and D-dimer measurements. One-way ANOVA with Tukey post hoc HSD test was applied to determine whether there are statistically significant differences between the levels of IL-6, PCT, CRP, hBD1, hBD2, WBC and Neu in groups of patients without cancer or infection, with infection but without cancer, with cancer but without infection and, finally, with both cancer and infection. Moreover, one-way ANOVA was used in the process of examining the relationship between the levels of hBD2 and hBD1 in bacterial, viral and fungal infections, respectively.

To compare the predictive potential of hDB2, CRP and PCT for inflammation of infectious etiology (as opposed to inflammation of non-infectious etiology), receiver operating characteristic (ROC) curves and the areas under the respective curve (AUC) were calculated. *p*-values less than 0.05 were considered statistically significant. The AUC was considered to be clinically relevant if it was ≥ 0.8 [42]. In addition, kinetic graphs of CRP and PCT were plotted.

## 3. Results

### 3.1. Patients’ Characteristics

A total of 423 serum samples from patients with inflammation of infectious and non-infectious etiology were included in the study. The mean age of the patients was 66 years and we received on average 3.7 serial samples from each patient (Table 1). Of them, 307 were in the group of non-infectious inflammations and 116 in the group of infectious inflammations. The mean age of each group was 65 and 69 years, respectively. The control group consisted of 46 healthy individuals with a mean age of 64 years (Table 1). The infection group was further subdivided according the site of the infection (Table 2).

### 3.2. hBD2 Is Increased in Patients with Inflammation of Infectious Etiology

We compared the levels of hBD1, IL-6, hBD2, CRP, PCT, WBC and Neu between the patients with inflammation of infectious and non-infectious etiology, respectively. Regarding hBD2, *t*-test analysis and one-way ANOVA revealed a significant difference (*p* < 0.0001, F = 58.3) between serum concentration of hBD2 in patients with inflammation of infectious etiology compared to patients with inflammation of non-infectious etiology (*p* < 0.0001, t = 10.17) and healthy individuals (*p* = 0.0001, t = 3.714). No statistically significant difference was observed in the levels of hBD1 between patients with inflammation of infectious and non-infectious origin and between patients with an infection and healthy individuals (Figure 1). In addition, we observed a statistical difference between serum concentration of IL-6, PCT, WBC and CRP in patients with inflammation of infectious etiology compared to patients with inflammation of non-infectious etiology (*p* < 0.001, t = 7.353; *p* < 0.013, t = 2.493; *p* = 0.049, t = 1.985; and *p* = 0.023, t = 3.065, respectively) (Figure 1 and Figure 2).

### 3.3. Predictive Ability of hBD2 for Infection

In order to analyze the diagnostic value of individual laboratory biomarkers in predicting infection, ROC analysis was performed for the following markers for inflammation of infectious origin: hBD2, CRP and PCT (Figure 3). hBD2 serves as an efficient marker for the discrimination between inflammation of infectious and non-infectious origin, showing the highest AUC (AUC: 0.898; 95% CI: 0.838 to 0.958; *p* < 0.0001). On the other hand, CRP and PCT did not serve as well as infection classifiers, with their AUC being non-statistically significant (0.576; 95% CI, 0.473 to 0.678; *p* = 0.157) and CRP (0.517; 95% CI, 0.411 to 0.623; *p* = 0.747) (Figure 3).

The optimal cutoff of hBD2 for efficient discrimination between infectious and non-infectious inflammation was determined as 655 pg/mL using Youden Index analysis of the ROC curves. Using this cutoff for hBD2 positivity, hBD2 served to classify the patients’ sera in two categories (inflammation of infectious and non-infectious origin) with a sensitivity of 78.0% and a specificity of 93.2%. The positive likelihood ratio (PLR) was 11.4, the positive predictive value (PPV) was 88.6%, the negative predictive value (NPV) was 86.1% and the accuracy of the test was 87.0%. Using the same procedure, PCT and CRP failed to effectively classify the inflammation regarding the presence of infection. Their PLR, PPV, NPV and accuracy were 2.19, 60%, 63.1% and 62.6% for PCT and 1.27, 46.6%, 74.3% and 54.5% for CRP, respectively.

### 3.4. hBD1 and hBD2 Levels in Patients with Cancer

One-way analysis of variance demonstrated that the levels of hBD2 and hBD1 in sera of patients with cancer (without infection) were not significantly higher compared to the levels in patients without cancer (without infection) or healthy individuals (Figure 4). However, statistically significant differences in the analysis of variance (F = 29.71, *p* < 0.001) were observed when an infection was present either in cancer or non-cancer patients compared to patients with the same diagnosis but without infection (Figure 4A). Similarly, the mean levels of hBD2 were significantly higher when infection was present either in cancer (t = 5.25, *p* < 0.0001) or non-cancer patients (t = 8.07, *p* < 0.0001) as compared with the absence of infection in the same categories. CRP levels were not statistically different between all categories of patients with inflammation (Figure 4B).

### 3.5. hBD1 and hBD2 Levels in Different Types of Infection

We assessed whether hBD1 and hBD2 levels were significantly higher in certain types of infections. We analyzed the results regarding the following types of infection: (a) tissue infections, (b) urinary tract infections, (c) gastrointestinal tract infections, (d) respiratory tract infections and (d) bacteremia (data not shown). No statistically significant correlation was found between hBD1 levels and any specific type of infection. On the other hand, hBD2 levels were higher in patients with pneumonia [followed, in some cases, by Acute Respiratory Distress Syndrome (ARDS)] as compared to the other types of infection (*p* < 0.0001, t = 4.98) (Figure 5A). CRP levels were not statistically different between patients with pneumonia/ARDS compared to patients with other types of infection (Figure 5B).

### 3.6. Serum hBD2 Significantly Correlates with the Levels of IL-6

Next, we evaluated the correlations amongst individual parameters in patients with inflammation regarding the levels of hBD1, hBD2, CRP, IL-6, PCT, WBC, Neu, HCT, PCT, TKE, Glucose, Urea, Creatinine, K^+^, Na^+^, TBIL, LDH, ALB, CPK, INR and D-dimers. Pearson’s analysis indicated that there is a significant association between the levels of IL-6 and hBD2 (r = 0.8681, *p* < 0.0001) (Figure 6A). There is also a weaker but statistically significant association between the levels of IL-6 and CRP (r = 0.1631, *p* < 0.0001). Furthermore, a weak correlation was found between the levels of IL-6 and hBD1 (*p* = 0.0246) and the levels of CRP and hBD2 (*p* = 0.0230) (Figure 6).

### 3.7. Kinetic Analysis of hBD2 and CRP in Patients’ Sera Collected at Different Timepoints

We analyzed sera collected at different time points, from the same patient, in order to identify trends in the kinetics of hBD2 and CRP levels during the six first days post inflammatory stimuli, as shown in Figure 7A. Inflammatory stimuli included acute myocardial infarction, invasive cancer, trauma, surgical procedure, autoinflammatory disease exacerbation, infection, etc., that was clinically correlated with the observed CRP elevation. In patients with an infection, baseline hBD2 levels remained increased during the first five days post infection and decreased on the sixth day. In contrast, in patients with inflammation of non-infectious origin, hBD2 was significantly lower during all the six first days of inflammation. On the other hand, CRP kinetics were similar without any significant difference in groups of patients with inflammation of infectious vs. non-infectious origin. CRP levels gradually decreased in both groups through the first six days post inflammation (Figure 7B).

### 3.8. Analysis of hBD2 Kinetics in Specific Cases

In order to study the kinetics of hBD2 (elevation, peaking and reduction) in response to inflammation stimuli of different intensity, we analyzed patient’s sera collected at different time points in specific cases.

Case#1, patient MK88: An 88-year-old Caucasian man with history of a renal mass in the right kidney hospitalized due to acute myocardial infarction. During his hospitalization, he presented pulmonary edema and acute renal failure and was admitted to the Intensive Care Unit (ICU) on mechanical ventilation. On day 1 and day 4, the patient received hemodialysis. Due to their molecular weight, CRP (118kD), IL-6 (21kD) and PCT (14.5kD) biomarkers could not pass through the dialysis membranes and had to be removed from the patient’s serum. Therefore, CRP, IL-6 and PCT peaked on day 2 and their levels gradually decreased on day 5. On the other hand, low-molecular-weight hBD2 (3.5 kD) was removed on day 1 and day 4. hBD2 secretion followed a linear increment pattern on day 2 and day 3, with a doubling time of 13.8 h. On day 4, hBD2 was removed (serum sample just before dialysis was not available) and its concentration dropped to the same levels that hBD2 had after hemodialysis on day 1. On day 5, hBD2 concentration increased again as a response to the remaining inflammatory stimulus (of lower intensity) (Figure 8).

Case#2, patient PS59: A 59-year-old female admitted to hospital for acute ischemic stroke (AIS) and submitted to basilar artery thrombectomy and stent-assisted recanalization. On day 8, a Pseudomonas aeruginosa catheter infection was reported. IL-6 peaked first on day 8, while CRP and hBD2 peaked two days later. Both CRP and hBD2 secretion followed a linear increment pattern on day 6 until day 10, with a doubling time of 9.4 h and 15.8 h, respectively (Figure 8).

Case#3, patient SP85: An 85-year-old female with renal insufficiency admitted to hospital with acute respiratory failure and then to the ICU on mechanical ventilation under broad-spectrum antibiotic therapy. After 20 days in the ICU (day 1 shown in the graph), Candida albicans was isolated in a blood culture. Candidemia triggered the elevation of IL-6, hBD2 and CRP levels on day 2. Following an anti-fungal therapy, the levels of all three biomarkers decreased gradually during day 2 and day 9. The patient died 12 days later (Figure 8).

## 4. Discussion

Inflammation is the body’s response to tissue damage induced by injury or pathogens. It can be divided into acute inflammation, which is a short-term response, occurring within minutes or days, and chronic inflammation, which is a long-term response. This persistent type of inflammation is manifested in various diseases, including cancer and autoimmune diseases. Inflammation can be triggered due to an infectious etiology (e.g., microbial or viral infection) or a non-infectious etiology (e.g., tissue damage, cancer, cardiovascular disease and oxidative stress). Inflammatory responses are characterized by the production of several molecules, such as acute-phase proteins (e.g., CRP), pro-inflammatory cytokines (e.g., IL-6) and antimicrobial peptides (e.g., human β-defensins). The production of CRP is non-specific and occurs during inflammation of both infectious and non-infectious etiology. This biomarker is used to provide clinically relevant information, such as screening of an organic disease or monitoring of response to treatment [43]. On the other hand, human β-defensins are a family of naturally derived antibiotic peptides produced by the human body in the case of an infection able to kill a wide range of microorganisms such as fungi, bacteria and viruses.

In the present study, we report for the first time that the levels of hBD2 in the serum of patients with an infection are elevated and statistically different compared to those in patients with inflammation of non-infectious etiology and healthy individuals. In addition, it was demonstrated (by ROC curve analysis) that hBD2 is a superior biomarker to both PCT and CRP for the diagnosis of infection. hBD2 showed significant diagnostic accuracy in patients with an infection, in contrast to PCT and CRP which, in our study, failed to discriminate an infectious from non-infectious origin of inflammation.

CRP is a very useful non-specific biochemical marker of inflammation used for diagnosis in every day clinical practice. However, CRP is not able to help efficiently distinguish inflammation of infectious from inflammation of non-infectious etiology. Another inflammatory biomarker that has been associated with infection is PCT. PCT is produced by thyroid and hepatic cells. Its concentration in plasma increases 3-6 h after the inflammatory stimulus is introduced and reaches its maximum within 12 h. This biomarker is used for the diagnosis of several serious/life-threatening infections, such as sepsis and bacterial meningitis [44,45,46]. In addition, it is useful for detecting invasive bacterial infections in febrile infants [47]. However, PCT has not been as widely used in diagnosis as CRP. Various efforts to identify new biomarkers for infection have been undertaken during the last decade without success.

The discovery of new biomarkers that can differentiate between inflammation of infectious vs. non-infectious etiology is critical, since they will allow for the prompt diagnosis of infection and antibiotic intervention. The symptoms of the inflammatory process (e.g., fever) and the main laboratory findings (e.g., leukocytosis, CRP and ESR increase) are common in many different types of inflammation. This does not allow the differentiation between the different causes of inflammation. Clinicians often face a dilemma, as they need to determine with certainty whether the inflammation observed is of infectious or non-infectious etiology. For example, is the extensive inflammation observed in the case of a burn victim as a result to the extensive tissue damage or the super-infection of the burns? In the case of a resected patient, what causes the inflammation: the wound, the contamination of the wound or a surgical complication with microbial dispersion? Is the inflammation observed in a patient of the ICU due to the Systemic Inflammatory Response Syndrome (SIRS) or sepsis? In a patient who receives immuno-modulatory therapy for rheumatoid arthritis, is the inflammation observed due to the arthritis or an infection caused by the immune-modulatory therapy? In a cancer patient receiving chemotherapy, is the inflammation the result of the tumor’s invasive spread or an infection caused by immunosuppression?

The discovery of new biomarkers would also result in a reduction in the total number of tests that should be conducted to reach an accurate diagnosis, as well as a reduction in unnecessary antibiotic use and hospitalization time. Timely and correct diagnosis will lead to the immediate administration of appropriate and targeted therapeutic regimens and, consequently, the reduction of mortality and hospitalization time. All of the above, will not only improve the patients’ quality of life, but will also contribute to the reduction of healthcare costs for the National Healthcare System.

Another finding of this study is the significant association between the levels of IL-6 and hBD2. Analysis of sera collected at different time points revealed that hBD2 (as CRP) peaked about 24 h later than IL-6 and, in most cases, IL-6 secretion precedes the hBD2 peak (or it takes place simultaneously in some cases). In addition, the results of our study are in concordance with a study which evaluated the expression of human beta-defensin 2, TNF alpha and interleukins 1 alpha, 6 and 8 in 14 skin biopsies of psoriatic lesions by immunohistochemical staining. A statistically significant correlation between human beta-defensin 2 and IL-6 was found in the skin biopsies [48]. Another study indicated that hBD2 is related to cytokine production. More specifically, it was found that hBD2 is able to stimulate the production of IL-6, IL-8 and IL-10 from human Peripheral Blood Mononuclear Cells (PBMCs) in a dose-dependent manner [49].

Our results demonstrated that hBD2 levels are higher in patients with pneumonia compared to the other types of infection. In contrast, hBD1 levels are not statistically different between patients with pneumonia and patients with other types of inflammation or the normal population. In line with these observations, previous in vitro studies demonstrated the induction of hBD2, but not hBD1, in lung epithelial cells by inflammatory cytokines [50]. Not only inflammatory lung diseases but also infectious lung diseases are characterized by a marked increase in defensin hBD2 levels [51]. In this regard, infection with L. pneumophila or S. pneumoniae markedly induced hBD2 expression in human pulmonary epithelial cells in a Toll-like receptor 2 (TLR2)/NF-κB-dependent manner [52]. Moreover, patients with Mycobacterium avium intracellular infection [53] and patients with acute bacterial pneumonia [54] show markedly elevated hBD2 (but not hBD1) levels in both plasma and bronchoalveolar lavage fluid (BALF). It has been proposed that hBD2 might play a role in the host defense and local remodeling of the respiratory tract in these patients. Song Liu et al. suggested that low hBD2 levels on admission are associated with 30-day adverse clinical outcomes in patients with community-acquired pneumonia (CAP) and that plasma hBD2 levels might be a useful tool for the prognostic stratification in patients with CAP [55]. Singh et al. also described increased β-defensin levels in bronchioalveolar lavage fluid (BALF) of cystic fibrosis (CF) patients. Interestingly, increased levels of hBD2 were observed, while hBD1 levels remained unchanged [39].

Furthermore, in our study it was shown that the levels of hBD2 and hBD1 in patients with different kinds of cancer (without infection) were not significantly higher compared to the levels in patients with inflammation without cancer and without infection or healthy individuals. It has been previously reported that the expression of hBD1 is deficient or reduced in many renal cell carcinomas, prostate cancers [56], basal cell carcinomas [57] and oral squamous cell carcinomas (OSCC) [58]. On the other hand, other studies supported the notion that hBD1 is increased in renal cell carcinomas [59]. In this regard, it was shown that the levels of hBD1 and hBD2 were elevated in the serum of patients with lung cancer [60]. It has been reported that hBD2 and hBD3 may have a role as proto-oncogenes in Oral Squamous Cell Carcinomas (OSCC), while hBD1 may act as a tumor suppressor [61]. Taking all the above into consideration, it is clear that the function and the mechanisms of the defensins’ expression in cancer are complex, not-yet-clarified processes.

Analysis of sera collected at different time points demonstrated that hBD2 helps differentiate among infectious and non-infectious etiology from the first day post inflammation. After the cure of the infection with antibiotic treatment, hBD2 concentration decreases on day 6 and returns to baseline levels several days later. In contrast, CRP only showed a tendency to slightly increase during the first 2 days post infection (as compared to inflammation of non-infectious origin) and could not help clearly differentiate between the two cases. The kinetics of hBD2 secretion in one case where hBD2 was removed by the patient’s blood via hemodialysis revealed that hBD2 increases rapidly, with a doubling time of 13.8 h. Data from other patients demonstrated that hBD2 followed the pattern of CRP from the time point of peaking and after the treatment of the infection. hBD2 returned to low levels with a sub-doubling time similar to the one of CRP. The fold increase in hBD2 was dependent on the intensity of the stimulus, but in most cases was similar to the one of CRP. Taken together, hBD2 increases rapidly, remains at high levels (up to 100×) for a few days and falls quickly to baseline levels once the infection subsides. Moreover, hBD2 can be easily detected by routine laboratory methods. These characteristics make hBD2 an ideal biomarker for the monitoring of infections and the efficacy of antibiotic therapy.

## 5. Conclusions

Taking all the evidence of our study into consideration, we conclude that hBD2 has the potential to serve as a novel diagnostic marker for infection. The use of hBD2 as a biomarker could provide an added value to the clinical decision-making process, assessing prognosis and assisting in diagnosis and treatment selection and monitoring.

## Figures and Tables

**Figure 1 diagnostics-13-01885-f001:**
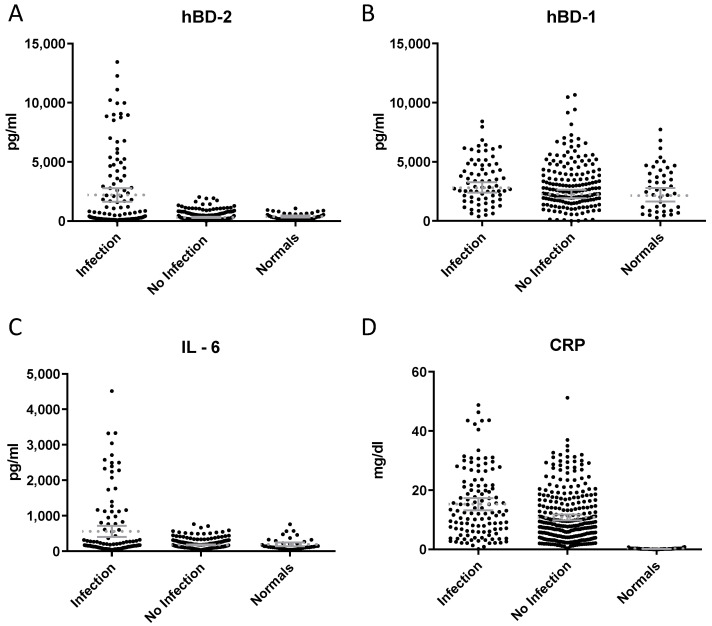
Serum levels of hBD2 (**A**), hBD1 (**B**), IL-6 (**C**) and CRP (**D**) in patients with inflammation of infectious/non-infectious etiology and healthy individuals. Patients with inflammation of infectious etiology had higher levels of hBD2 (*p* < 0.0001, t = 10.17), IL-6 (*p* < 0.001, t = 7.353) and CRP (*p* = 0.023, t = 3.065) as compared with patients with inflammation of non-infectious etiology. No statistical difference was observed in the levels of hBD1 between patients with inflammation of infectious and non-infectious origin (*p* = NS, t = 1.180).

**Figure 2 diagnostics-13-01885-f002:**
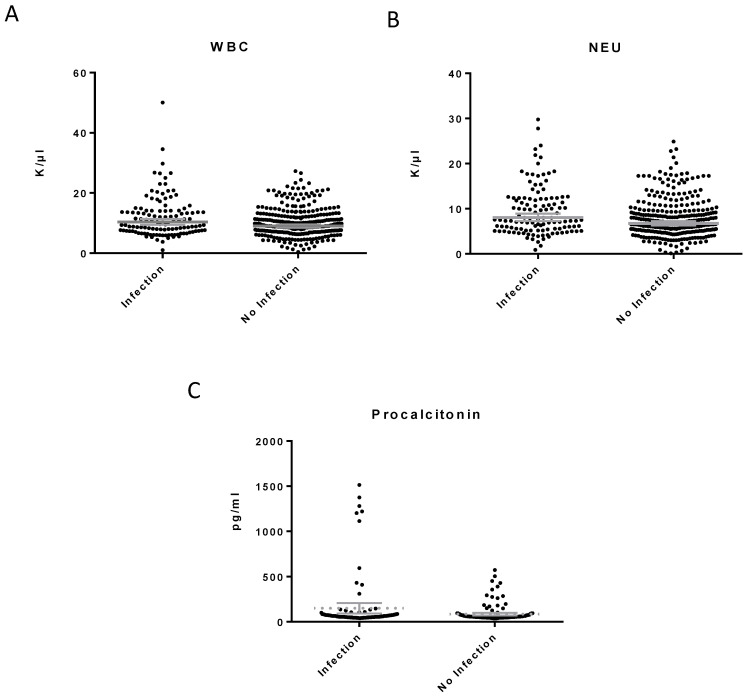
White blood cell (WBC) (**A**), neutrophil (NEU) (**B**) and procalcitonin levels (**C**) in patients with inflammation of infectious and non-infectious etiology. Patients with inflammation of infectious etiology had marginally higher levels of WBC (*p* = 0.049, t = 1.985), Neu (*p* = 0.030, t = 2.169) and procalcitonin (*p* = 0.013, t = 2.493) as compared with patients with inflammation of infectious etiology.

**Figure 3 diagnostics-13-01885-f003:**
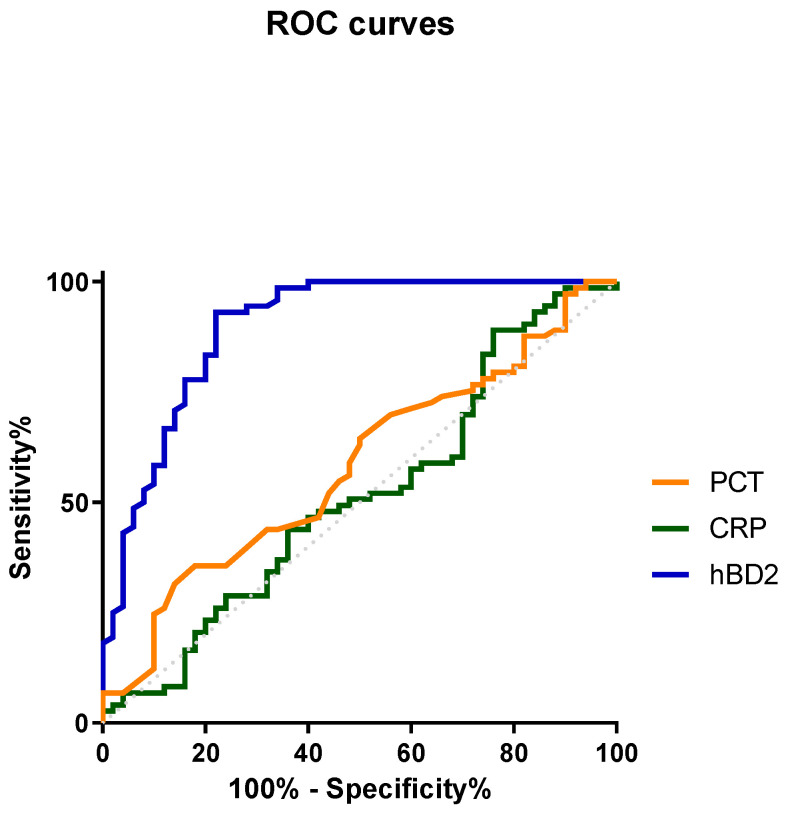
ROC analysis for hBD2 (0.898; 95% CI, 0.838 to 0.958; *p* < 0.0001), PCT (0.576; 95% CI, 0.473 to 0.678; *p* = 0.157) and CRP (0.517; 95% CI, 0.411 to 0.623; *p* = 0,747) as classifiers for infectious inflammation.

**Figure 4 diagnostics-13-01885-f004:**
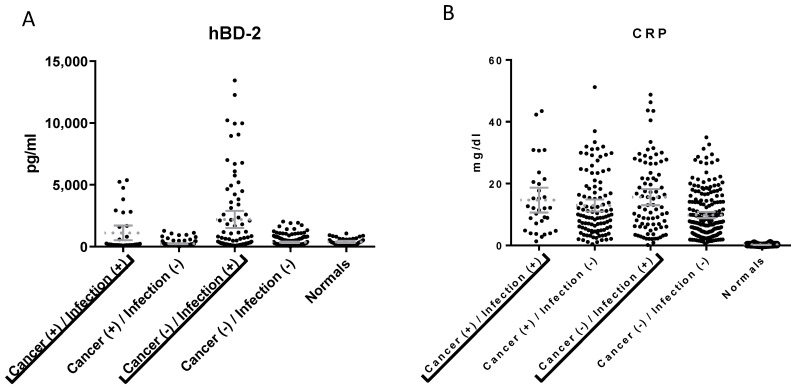
Serum levels of hBD2 (**A**) and CRP (**B**) in patients with cancer (with or without infection) compared to patients without cancer (with or without infection) and healthy individuals (left). hBD2 levels were higher in patients with cancer and infection as compared to patients with cancer without infection (t = 5.25, *p* < 0.0001) and in non-cancer patients with infection as compared to non-cancer patients without infection (t = 8.07, *p* < 0.0001). CRP levels were not statistically different between all categories of patients with inflammation (with or without cancer).

**Figure 5 diagnostics-13-01885-f005:**
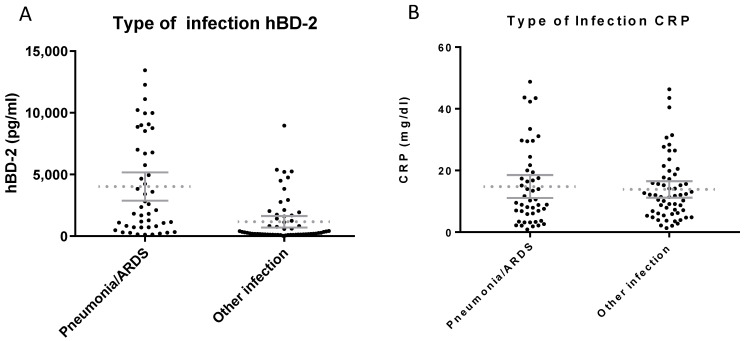
Serum levels of hBD2 (**A**) and CRP (**B**) in patients with pneumonia or another type of infection. hBD2 levels were higher in patients with pneumonia/ARDS as compared to the other types of infection (*p* < 0.0001, t = 4.98) while CRP levels were not statistically different between the same groups of sera.

**Figure 6 diagnostics-13-01885-f006:**
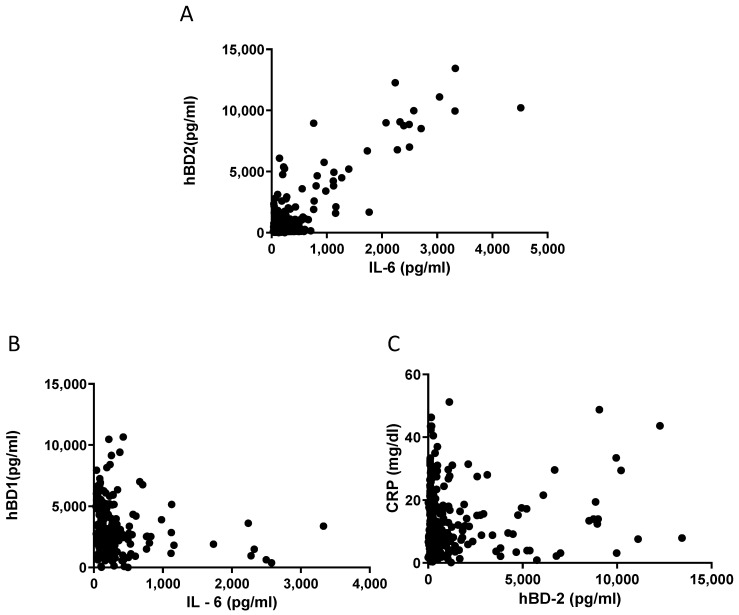
(**A**) hBD2 concentration is correlated with the levels of IL-6. Pearson’s analysis r = 0.8681, *p* < 0.0001. (**B**) hBD1 concentration is correlated with the levels of IL-6. Pearson’s analysis *p* = 0.0246. (**C**) hBD2 concentration is correlated with the levels of CRP. Pearson’s analysis *p* = 0.023.

**Figure 7 diagnostics-13-01885-f007:**
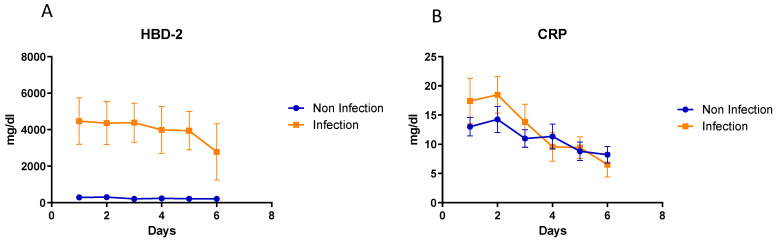
Kinetics in serial sera of the mean hBD2 (**A**) and CRP (**B**) levels in patients with infectious versus non-infectious inflammation.

**Figure 8 diagnostics-13-01885-f008:**
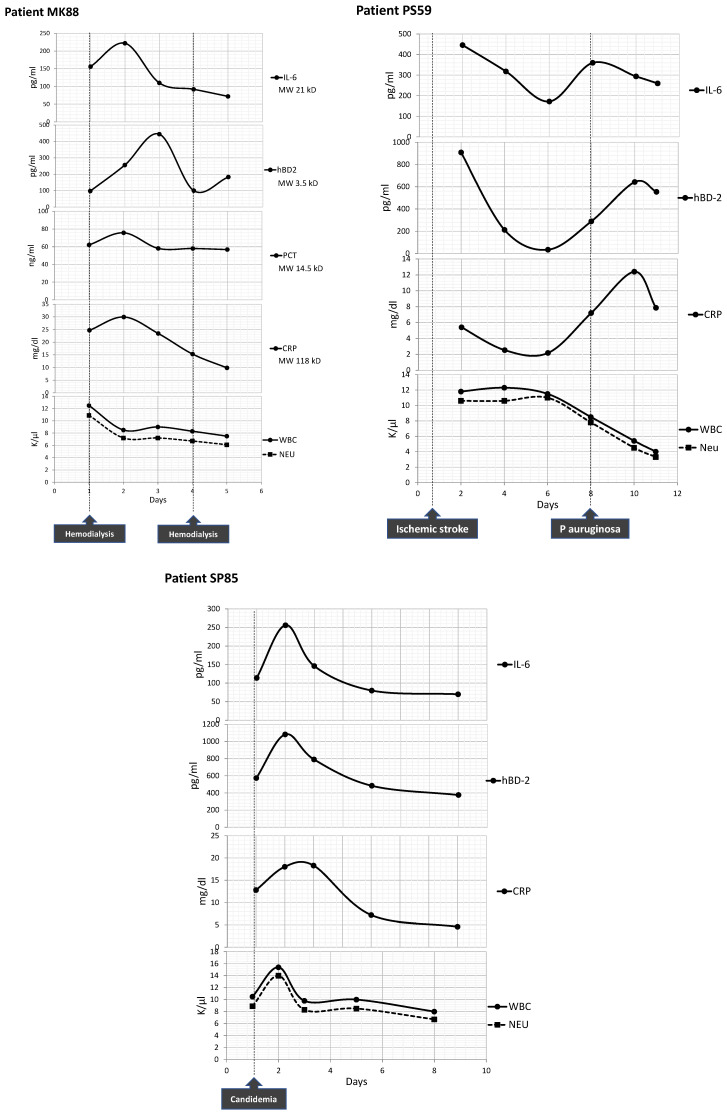
Analysis of IL-6, hBD2, PCT, CRP, WBC and Neu kinetics for patients MK88, PS59 and SP85.

**Table 1 diagnostics-13-01885-t001:** Features of the study population.

Variables	Patients	Infection	No Infection	Healthy Individuals
Number of individuals	114	42	72	46
Number of samples	423	116	307	46
Serial samples/patient	3.7	2.8	4.3	N/A
Age (years) ± SD	66 ± 18	69 ± 3	65 ± 2	64 ± 20
Female/Male	68/46	25/17	42/30	28/18
CRP (mg/dL) ± SD	12.1 ± 9.6	14.3 ± 11.4	11.3 ± 8.5	0.2 ± 0.16
hBD1 (pg/mL) ± SD	2942 ± 1876	3337 ± 1884	3059 ± 1810	2858 ± 2020
hBD2 (pg/mL) ± SD	853.1 ± 1018	2304.0 ± 2071	311.9 ± 247.9	378.2 ± 281.1
PCT (ng/mL) ± SD	112.8 ± 165.3	150.6 ± 197.0	86.8 ± 90.0	0.6 ± 0.7
WBC (K/μL) ± SD	10.7 ± 5.6	11.4 ± 6.7	10.4 ± 4.8	6.3 ± 3.5
NEU (K/μL) ± SD	8.4 ± 4.8	8.9 ± 5.4	8.2 ± 4.4	3.9 ± 2.7

N/A: not applicable.

**Table 2 diagnostics-13-01885-t002:** Types of infection.

	Number of Patients	Number of Samples
Urinary tract infection	9	20
Wound/tissue infection	7	23
Respiratory infection	12	37
Blood infection	5	9
Sputum infection	3	6
Gastrointestinal infection	4	18
Other types of infection	2	3
Infection total	42	116

## Data Availability

Not applicable.

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
