# Peer review of "Serum β-Defensin 2, A Novel Biomarker for the Diagnosis of Acute Infections"

_diagnostics, 2023, doi:10.3390/diagnostics13111885_

Round 1

Reviewer 1 Report

Routsias JG et al propose serum beta-defensin2 as a novel biomarker of acute infection. Their paper offers a comprehensive review on the subject of acute inflammation biomarkers before presenting their data, as a welcome addition. 
It is clearly stated that serum beta-defensin 2 peaks 24 hours after IL6, but it is not explained why dosing serum beta-defensin2 should be preferred to the one of IL6: is it there any discrepancy between serum beta-defensin2 and IL6? or the range values of serum beta-defensin2 amplify much better the ones of IL-6 or something else?
The three cases (MK88, PS59 and SP85) presented should show PCT values even for PS59 and SP85 and not only for MK88; it would be interesting to know how it is serum beta-defensin2 in a case of viral infection like influenza or Covid-19 where we know there is discrepancy between PCR and PCT.
Last but not least, PCT is often used as a help to decide to stop antibiotic therapy when PCT normalizes; can this study suggest something similar for serum beta-defensin2?

Author Response

We thank the reviewer for their helpful suggestions. The comments were addressed as follows:

Point 1. It is clearly stated that serum beta-defensin 2 peaks 24 hours after IL6, but it is not explained why dosing serum beta-defensin2 should be preferred to the one of IL6: is it there any discrepancy between serum beta-defensin2 and IL6? or the range values of serum beta-defensin2 amplify much better the ones of IL-6 or something else?

Response 1: The half-life of Il-6 in serum is very short (ie https://doi.org/10.1084/jem.169.1.333, https://doi.org/10.1002/jcla.22924) and thus it is eliminated in a short time. Moreover, elevated proinflammatory cytokine levels (such as IL-6) usually precede symptoms onset in patients with inflammation (ie https://doi.org/10.1097/CCM.0000000000001188). These two characteristics are unpreferable for the use of IL-6 as a diagnostic marker. On the other hand, hBD2 increases rapidly in infection, remains in high levels (up to 100x) for a few days and falls quickly to baseline levels once the infection subsides. Data from patients demonstrated also that hBD2 followed the pattern of CRP elimination from the time point of peaking and after the treatment of the infection.  

Point 2. The three cases (MK88, PS59 and SP85) presented should show PCT values even for PS59 and SP85 and not only for MK88

Response 2: Unfortunately, we do not have PCT data for different time points for patients PS59 and SP85 to present their PCT data kinetics. We have PCT data only for 1 or 2 time points that are insufficient for a graph representation.  For the consistency of graphs in Figure 8, one option is to remove PCT from patient’s MK88 graph, if the reviewer suggests so.

Point 3. Last but not least, PCT is often used as a help to decide to stop antibiotic therapy when PCT normalizes; can this study suggest something similar for serum beta-defensin2?

Response 3: Yes, as it is stated in the "Discussion" section, our data suggests that after infection onset hBD2 increases rapidly, remains in high levels (up to 100x) for a few days and falls quickly to baseline levels once the infection subsides. This is very helpful to make the decision to stop the antibiotic course.

Reviewer 2 Report

Good manuscript with minor revisions required.  

Author Response

We thank the reviewer for the helpful suggestions. The comments were addressed as follows:

Point 1: Introduction: Improve the presentation of the problem.

Response 1:  The manuscript's section was edited according the reviewer’s suggestions (see attachment).

Point 2: Improve this section (Study Design and Participants). Selections criteria are not clear. 

Response 2: The manuscript's section was edited according the reviewer’s suggestions (see attachment).

Point 3: Materials and Methods: There is lack of references in this section.

Response 3:  For laboratory measurements, we used commercial assays (and not in-house methods). Thus, there are no references to add. For each commercial assay the manufacturer’s name is given ig Peprotech Inc, for hIL-6, hBD1 and hBD2, RayBiotech Inc, for PCT assay etc.

Point 4: Table 1. Features of the study population: Give the SD for all variales. 

Response 4:  The manuscript's Table 1 was edited according the reviewer’s suggestions (see attachment).

Point 5: Figures 1-5: Insert the results of statistical analysis in the figures. 

Response 5:  The manuscript's legends of the figures 1-5,  were edited according the reviewer’s suggestions (see attachment).

Point 6: Discussion: Make comparison with other works.  

Response 6: In the “Discussion” section we have compared our results with other works as it is stated in lines 368-381: “In line with these observations, previous in vitro studies demonstrated the induction of hBD2, but not hBD1, in lung epithelial cells by inflammatory cytokines [50]. Not only inflammatory lung diseases, but also infectious lung diseases are characterized by a marked increase in defensin hBD2-levels [51]. In this regard, infection with L. pneumophila or S. pneumoniae markedly induced hBD2 expression in human pulmonary epithelial cells in a Toll-like receptor 2 (TLR2)/NF-κB-dependent manner [52]. Moreover, patients with Mycobacterium avium intracellular infection [53] and patients with an acute bacterial pneumonia [54] show markedly elevated hBD2 (but not hBD1) levels in both plasma and bronchoalveolar lavage fluid (BALF). It has been proposed that hBD2 might play a role in the host defense and local remodeling of the respiratory tract in these patients. Song Liu  et al suggested, that low hBD2-levels on admission are associated with 30-day adverse clinical outcomes in patients with community-acquired pneumonia (CAP) and that plasma hBD2-levels might be a useful tool for the prognostic stratification in patients with CAP [55]. Singh et al. also described increased β-defensin levels in bronchioalveolar lavage fluid (BALF) of cystic fibrosis (CF) patients. Interestingly, were observed increased levels of hBD2, while hBD1 levels remained unchanged [39].”

Reviewer 3 Report

The Authors presented a very interesting article concerning the research and the possible use of β-defensin 2 as a new clinical marker for the evaluation of acute infection.

The document is very well designed and easy to read. The results are clearly present, but, we would like to suggest to the author, the possibility of presenting the graphs with colors (there are many gray scales).

Patient categories are very well selected, especially those representing cancers and infectious diseases. In fact, the authors have adequately discussed the problems related to this aspect.

The conclusions are appropriate in accordance with the results obtained and presented.

We would like to stimulate the discussion of the authors again with the following indications, in order to increase the solidity of the manuscript.

Minor revisions:

1) BD2 appears to be associated (especially) with lung lesions.

Do the authors have any data regarding yeast infection in lung patients (with or without cancer)?

2) Probably (it is not mandatory), but the reproduction of color graphics could increase the visibility of paper.

The document presented in its current form is suitable for publication.

Only two minor questions were addressed to the authors, but they are not mandatory for the final decision.

Best regards

Author Response

We thank the reviewer for the helpful suggestions. The comments were addressed as follows:

Point 1: BD2 appears to be associated (especially) with lung lesions. Do the authors have any data regarding yeast infection in lung patients (with or without cancer)?

Response 1: There were 5 patients with yeast infection (Candida) in lung, 1 with cancer and 4 without cancer, but the sample size was not big enough for statistical analysis and graph representation.

 Point 2: Probably (it is not mandatory), but the reproduction of color graphics could increase the visibility of paper.

Response 2: Figures 3 and 7 were prepared in color graphics, according to the reviewer’s suggestion (please see attachment).
